# Histopathological Evaluation of *Annona muricata* in TAA-Induced Liver Injury in Rats

**Morteta H. Al-Medhtiy [1], Ahmed Aj. Jabbar [2], Suhayla Hamad Shareef [3,4,\*], Ibrahim Abdel Aziz Ibrahim [5], Abdullah R. Alzahrani [5] and Mahmood Ameen Abdulla [6]**

[1]  Department of Anatomy and Histology, Faculty of Veterinary Medicine, University of Kufa, Kufa 54003, Iraq
[2]  Department of Medical Laboratory Technology, Erbil Technical Health and Medical College, Erbil Polytechnic University, Erbil 44001, Iraq
[3]  Department of Biology, College of Education, Salahaddin University-Erbil, Erbil 44001, Iraq
[4]  Department of Medical Biochemical Analysis, College of Health Technology, Cihan University-Erbil, Erbil 44001, Iraq
[5]  Department of Pharmacology and Toxicology, Faculty of Medicine, Umm Al-Qura University, Makkah 77207, Saudi Arabia
[6]  Department of Medical Microbiology, College of Science, Cihan University-Erbil, Erbil 44001, Iraq
\*  Correspondence: suhayla.shareef@su.edu.krd; Tel.: +964-07728414580

**Abstract:** This research in vivo assessed the impact of the ethanolic extract of *Annona muricata* (*A. muricata*) on the histopathology, immunohistochemistry, and biochemistry of thioacetamide (TAA)-induced liver cirrhosis in *Sprague Dawley* rats. The rats, gavaged precisely with two doses of *A. muricata* (250 mg/kg and 500 mg/kg) with TAA, presented a substantial reduction in the liver index and hepatocyte propagation, with much lower cell injury. These groups showed meaningfully down-regulated proliferating cell nuclear antigen (PCNA) in the liver and spleen, α-smooth muscle actin (α-SMA), and transforming growth factor-beta 1 (TGF-β1) in liver parenchymal tissue. The liver homogenate displayed enhanced antioxidant enzymes, superoxide dismutase (SOD) and catalase (CAT) activity, along with a decrease in malondialdehyde (MDA) levels. The serum levels of bilirubin, total protein, albumin, and liver enzymes alkaline phosphatase (ALP), alanine aminotransferase (ALT), and aspartate aminotransferase (AST) were returned to normal and were similar to that of the normal control and silymarin with TAA-treated groups. Oral acute toxicity revealed no evidence of any toxic symbols or mortality in rats, indicating the safety of *A. muricata*. Therefore, the normal microanatomy of hepatocytes, the clampdown of PCNA, α-SMA, TGF-β, improved antioxidant enzymes (SOD and CAT), and condensed MDA with repairs of liver biomarkers validate the hepatoprotective effect of *A. muricata*.

**Keywords:** liver cirrhosis; TAA; antioxidants; histology; immunohistochemistry

## 1. Introduction

There is an upsurge in liver fibrosis prevalence worldwide, which may be caused by hepatitis B virus infection, the hepatoxic effects of chemicals such as alcohol, metabolic-associated fatty liver disease, medications, and environmental pollutants. Medicinal plants have been used conventionally for centuries for the treatment of liver disorders in humans. In the scientific literature, many studies have established the advantageous result of countless medicinal plants in protecting the liver, contrary to thioacetamide (TAA)-influenced damage in laboratory animals [1–6].

*Annona muricata* (Linn.) belongs to the Annonaceae family. It is a remedial plant that has been used as a natural therapy for diverse illnesses including liver diseases. A number of studies confirmed that *Annona muricata* leaves possess anti-ulcer [7], anti-cancer [8,9], anti-hypertensive [10], anti-diabetic [11], wound-healing [8], anti-bacterial [12], cytotoxic [13], and anti-proliferative activities [14]. The importance of *A. muricata* is likewise

donated to its numerous applications in traditional remedies [15]. All parts of *A. muricata*, counting the bark, fruit seeds, leaves, and root are utilized in usual medications in the tropics [16]. Conventionally, the leaves are utilized for cystitis, diabetes, pains, asthma, cough, fever, sedative, toothache, high blood pressure, sleeplessness, liver complications, and as an anti-dysenteric and spasmolytic [16,17]. The leaves of *A. muricata* have been found to have substantial antioxidant properties [18]. Furthermore, the leaves established a distinguished defensive influence against acute as well as chronic inflammations in rats, through the clampdown of pro-inflammatory cytokines [19].

Milk thistle (*Silybum marianum*) fruits comprise a mixture of flavonolignans, collectively recognized as silymarin. Silymarin is a flavonolignan that has been used recently as a hepatoprotective agent. Silymarin is used for the treatment of a variety of liver disorders categorized by degenerative necrosis and functional impairment [20]. Numerous studies by different scholars used silymarin as the usual drug for hepatoprotection against poisoning by thioacetamide [21–27]. Hepatotoxins primarily harm the liver cells wherever an additional amount of cytochrome P450 oxidases facilitate its modification to noxious intermediates, followed by an increase in reactive oxygen species, MDA, and pro-inflammatory cytokines [28]. A rise in free radicals harms proteins, adipocytes, and the DNA of nuclei [29]. TAA leads to hepatic cell injury after its breakdown to TAA sulphene and sulphone, by a serious trail that includes CYP4502E1-facilitated bio transdevelopment [30]. Some studies have proved that TAA is involved in the initiation of liver fibrosis [6,23,31–34]. The objective of the current study is to evaluate the histology and immunohistochemistry of the hepatoprotective effect of *A. muricata* against TAA-induced liver injuries in experimental rats.

## 2. Materials and Methods

### 2.1. Morals Declaration

This study was permitted through the Conscience Commission for Animal House Investigation, Faculty of Medicine, University of Malaya, Malaysia (Ethic No. PM 3 July 2020 MMA (a) (R)). All rats obtained humanitarian attention according to the principles described in the Guide for the Maintenance and Usage of Workroom Animals prearranged by the United States Nationwide College of Disciplines and made available by the Countrywide Institutions of Healthiness [35].

### 2.2. Thioacetamide

Thioacetamide (TAA) was acquired from Sigma-Aldrich, Switzerland, and dissolved in sterile 10% Tween 20, mixing well until completely liquified. At that same time, 200 mg/kg of TAA was injected i.p. to rats three times per week for 2 months. The TAA-induced changes in both biological and morphological features were similar to that of human liver fibrosis [36].

### 2.3. Silymarin

Silymarin (International Laboratory, San Francisco, CA, USA) is a reference medication dissolved in sterile 10% Tween 20 for oral feeding to animals at a dosage of 50 mg/kg body mass daily for two months [37].

### 2.4. Plant Preparations and Extraction

Fresh leaves of the *A. muricata* plant were obtained from the garden of Rimba Ilmu and identified via contrast with the voucher sample at the Herbarium of Rimba Ilmu, Institute of Science Biology (Voucher No. KLU47978). The plant's leaves were dried in shade and ground into fine powder. The leaf powder (500 g) was soaked in 2000 mL of 95% alcohol at 25 °C for three days. The combination was separated by using Whatman No. 1 paper and concentrated to dryness using a rotary vacuum evaporator. The excerpt was liquified in 10% Tween 20 and fed to animals at a dose of 250 mg/kg and 500 mg/kg (5 mL/kg)

once daily for two months based on the experimental proposal as reported by a previous study [2].

### 2.5. Experimental Animals for Liver Cirrhosis

Adult male rats (150–175 g/body weight) were acquired from the Experimental Animal House Unit and reserved in polypropylene cages (6 in each cage). Animals were allowed admission to water and normal laboratory pellets. Rats acclimatized under typical workroom circumstances for 5 days before starting the experiment.

### 2.6. Hepatoprotective Activity

Thirty healthy adult male rats were arbitrarily divided into 5 sets with 6 rats each.

Group 1 (normal healthy group) rats were fed by mouth with 10% Tween 20 daily and injected i.p. with distilled water 3 times per week for 2 months.

Group 2 (hepatotoxic group) rats were gavaged with 10% Tween 20 and injected i.p. with TAA (200 mg/kg) 3 times weekly for 2 months.

Group 3 (silymarin-fed group) rats were fed per os with silymarin (50 mg/kg) every day and injected i.p. with TAA (200 mg/kg) 3 times weekly for 2 months.

Group 4 and 5 (*A. muricata* fed groups) rats were fed po with the extract 250 mg/kg and 500 mg/kg day-to-day, respectively, and injected i.p. with TAA (200 mg/kg) 3 times weekly for 2 months [38].

The body masses of rats in entire groups were assessed every week. At the end of 2 months, the rats were fasted and sacrificed under general anesthesia using Ketamine and Xylazine [39]. Blood was collected from intracardial puncture for liver function markers. Part of the liver was placed in 10% buffered formalin for histology in addition to immunohistochemistry staining and the other part was homogenated for measurement of endogenous enzyme activities (superoxide dismutase (SOD) and catalase (CAT)) and lipid peroxidation (malondialdehyde (MDA)) levels.

### 2.7. Experimental Animals for Acute Toxicity Study

The examination of the safety of the plants prior to performing any experiment on their biological activity is highly suggested to avoid any unwanted adverse side effects. Male and female SD rats aged 5–7 weeks and with weights of 170–190 g was acquired from the Animal Experimental Unit. The rats were given normal rat pellets ad libitum and tap water. An acute toxic trial was practiced to fix a harmless dosage for the *A. muricata* plant. A total of 36 rats (18 males and 18 females) were allocated into 3 groups, categorized as vehicle (10% Tween 20), 2000 mg/kg, and 5000 mg/kg of *A. muricata* respectively. Experimental rats fasted overnight (allowed free access to water) prior to treatment. Animals were observed for 30 min and at 2, 5, 24, and 48 h for any toxic signs, abnormal behaviors, or death. Afterward, rats fasted overnight on the 14th day and were sacrificed on the 15th day using general anesthesia, Ketamine (30 mg/kg, 100 mg/mL), and Xylazine (3 mg/kg, 100 mg/mL) [39]. Gross necropsies and histopathology were performed on livers and kidneys, and blood biochemical parameters were measured following usual procedures as described by a previous study [40].

### 2.8. Gross Appearance of Liver

Livers were excised and promptly rinsed in cold phosphate buffer saline, blotted on filter paper, weighed using an electrical digital balance, and carefully inspected for any gross pathological changes by taking images [32].

### 2.9. Histopathology of Liver Tissue

Hepatic tissue slices were placed in freshly prepared 10% phosphate-buffered formalin (PBF) overnight for complete fixation, trimmed to small parts, re-fixed in PBF, and embedded in paraffin using a mechanical processing machine. Slices at 5 μm thickness were stained with H&E [39] and Masson trichrome stains [41] for histopathological assessment.

### 2.10. Immunohistochemical Staining

Poly-L-lysine-treated glass slides were used for proliferating cell nuclear antigen (PCNA), transforming growth factor-beta 1 (TGF-β1), and α-smooth muscle actin (α-SMA) staining methods, as previously described in detail [22,34].

### 2.11. Liver Tissue Homogenate for Endogenous Enzymes

A lobe of liver was rinsed instantly with cold PBS and normalized in cold PBS pH 7.2 by a Teflon homogenizer. The remaining tissues were divided by centrifuge at 3500 rpm for 15 min at $-4\ ^{\circ}$C. Supernatants were analyzed for the activity assessment of SOD, CAT, and MDA levels (Cayman Chemical Company, Ann Arbor, MI, USA) based on the manufacturer's protocol (Cayman Chemical Company, Ann Arbor, MI, USA) [42].

### 2.12. Biochemical Parameters

After two months, rats were fasted overnight and sacrificed under general anesthesia using Ketamine and Xylazine [40]. Blood was collected through an intracardiac puncture, placed in gel-activated tubes, centrifuged, and examined at the Clinical Diagnosis Laboratory of the University Malaya Hospital for liver biomarker parameters [43].

### 2.13. Statistics

Data were analyzed using one-way analysis of variance (ANOVA) through post hoc assessment by Tukey multiple judgments in the PASW database (version 25) for Windows (SPSS Inc. Chicago, IL, USA). The values were designated as mean $\pm$ S.E.M.; a result worth $p < 0.05$ was considered significant.

## 3. Results

### 3.1. Acute Toxicity Trial

Rats fed by *A. muricata* at a dose of 2 g/kg and 5 g/kg were observed for 2 weeks. Rats were continuously active and there were no significant signs of toxicity, atypical symbols, alteration of behavioral patterns, body mass deviations, or gross discovery noticed throughout the entire experiment. There was no mortality in both dosages towards the end of 14 days. A histopathological investigation of the liver and kidney and blood biochemical markers showed no substantial alterations among the various groups (Figure 1 and Table 1. Based on the findings of the current study, *A. muricata* was not toxic at either dosage.

**Table 1.** (**A**) Acute toxicity test: Influences of *Annona muricata* extract on liver biochemical markers in rats. (**B**) Acute toxicity test: Effects of *Annona muricata* extract on kidney biochemical parameters in rats.

| (A) | | | | | |
|---|---|---|---|---|---|
| **Animal Groups** | **ALP (IU/L)** | **ALT (IU/L)** | **AST (IU/L)** | **T. Bilirubin (µmol/L)** | **T. Protein (g/L)** | **Albumin (g/L)** |
| Normal control 10% Tween 20 | $73 \pm 2.89$ [a] | $38 \pm 2.36$ [b] | $58 \pm 2.36$ [c] | $1.12 \pm 0.23$ [d] | $70 \pm 3.22$ [a] | $25 \pm 1.41$ [e] |
| *A. muricata* 2 g/kg | $68.16 \pm 2.83$ [a] | $40.16 \pm 2.85$ [b] | $61.16 \pm 2.31$ [c] | $1.15 \pm 0.04$ [d] | $68.20 \pm 3.71$ [a] | $23.34 \pm 2.22$ [e] |
| *A. muricata* 5 g/kg | $74 \pm 3.40$ [a] | $37 \pm 1.78$ [b] | $57 \pm 1.75$ [c] | $1.11 \pm 0.02$ [d] | $73.16 \pm 2.31$ [a] | $22.16 \pm 2.29$ [e] |

| (B) | | | | |
|---|---|---|---|---|
| **Animal Groups** | **Sodium mmol/L** | **Potassium mmol/L** | **Chloride mmol/L** | **Urea mmol/L** | **Creatinine µmol/L** |
| Normal control 10% Tween 20 | $140 \pm 3.46$ [a] | $4.9 \pm 0.28$ [b] | $105 \pm 5.059$ [b] | $4.16 \pm 0.10$ [a] | $40.83 \pm 3.76$ [a] |
| *A. muricate* 2 g/kg | $144.16 \pm 3.65$ [a] | $5.01 \pm 3.65$ [b] | $135.5 \pm 46.98$ [a] | $5.02 \pm 0.32$ [a] | $38.36 \pm 2.95$ [a] |
| *A. muricate* 5 g/kg | $139.16 \pm 4.26$ [a] | $5.12 \pm 0.30$ [b] | $98.16 \pm 3.71$ [b] | $4.93 \pm 0.47$ [a] | $40.63 \pm 2.30$ [a] |

(**A**) Data were expressed as mean $\pm$ S.E.M There is no statistically important alteration between different groups. A similar letter within the same column means not significant at $p < 0.05$. (**B**) Values were stated as mean $\pm$ S.E.M. There are no statistically significant changes among diverse groups. Different letters within the same column are considered significant at $p < 0.05$.

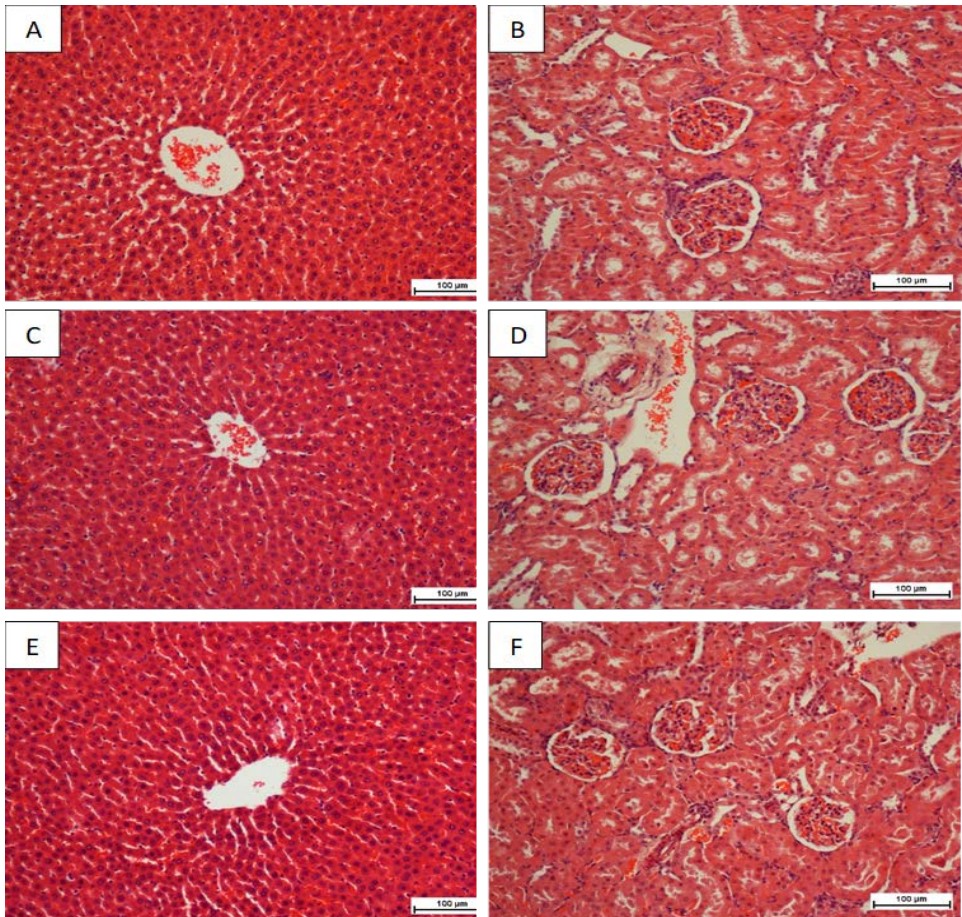

**Figure 1.** Effect of *A. muricata* extract on histological sections of the liver (**1st column**) and kidney (**2nd column**) from acute toxicity test in rats. (**A,B**) Normal control group, (**C,D**) rats fed with 2 g/kg of *A. muricata* extract, (**E,F**) rats fed with 5 g/kg of *A. muricata* extract. There is no significant alteration in the architecture of the livers and kidneys between the treated and control groups (Hematoxylin and Eosin stain, 20x).

### 3.2. Effect of A. muricata in Thioacetamide (TAA) Produced Liver Fibrosis

3.2.1. Body Weight, Liver Mass, and Liver Index

The weight of rats in the TAA control group was significantly lower and they showed increased liver bulk in contrast to the normal control group. Rats fed with silymarin or *A. muricata* expressively revealed improvements in body mass and a decline in their liver weight (Table 2).

**Table 2.** Effect of *A. muricata* on body weight, liver weight, and liver index in TAA-induced liver injury in rats after 2 months.

| Animal Groups | Body Weight (g) | Liver Weight (g) | Liver Index (%) LW/BW% |
|---|---|---|---|
| Normal control | $335.17 \pm 7.35$ [a] | $10.15 \pm 0.04$ [a] | $33 \pm 0.65$ [a] |
| TAA control (200 mg/kg) | $167.91 \pm 3.11$ [d] | $12.53 \pm 0.05$ [b] | $14 \pm 0.82$ [d] |
| Silymarin (50 mg/kg) + TAA | $317.37 \pm 5.53$ [b] | $10.26 \pm 0.08$ [a] | $31 \pm 0.63$ [b] |
| *A. muricata* (250 mg/kg) + TAA | $245.31 \pm 3.83$ [c] | $10.31 \pm 0.05$ [a] | $24 \pm 0.41$ [c] |
| *A. muricata* (500 mg/kg) + TAA | $268 \pm 4.30$ [c] | $10.55 \pm 0.03$ [a] | $26 \pm 0.55$ [c] |

Mean value $\pm$ SEM ($n = 6$). Values indicated by different superscripts within the same column are significantly different according to Tukey's honestly significant difference test at a $p < 0.05$ significance level.

### 3.2.2. Gross Appearance of Liver

The macroscopic appearance of the hepatotoxic (thioacetamide (TAA) control) liver presented coarse, uneven granular surface with micro-nodules. Silymarin and *A. muricata*-fed rats exhibited nearly smooth and even surfaces without micronodules (Figure 2). The normal control group had even and smooth-surfaced livers. Rats fed by silymarin or *A. muricata* remarkably enhanced the recovery of liver structure from impairment caused by TAA and protected the liver from additional corrosion.

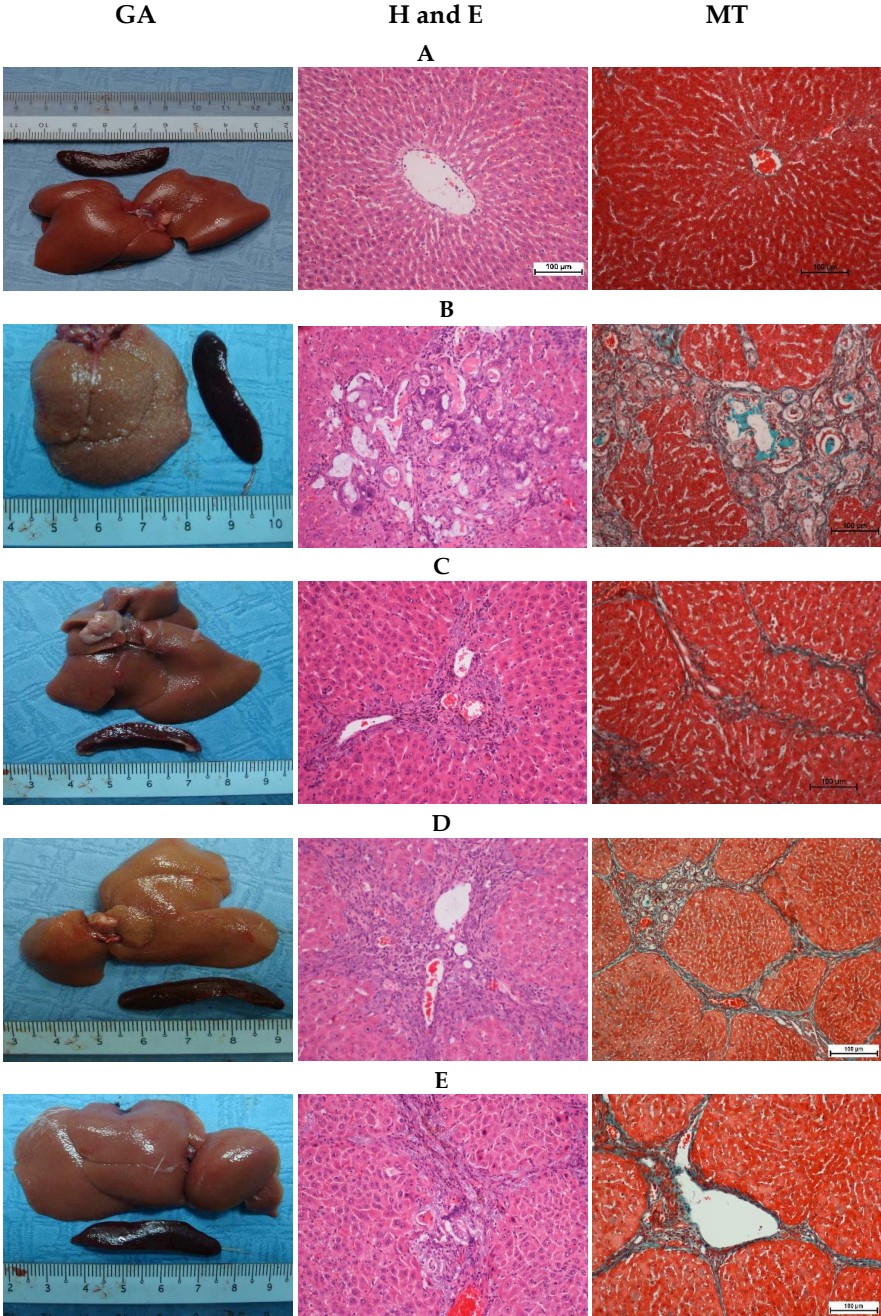

**Figure 2.** Influences of *A. muricata* extract on thioacetamide-induced liver injury in experimental rats. (**A**) Normal control group, (**B**) rats treated with thioacetamide (hepatotoxic group), (**C**) rats treated with thioacetamide + silymarin, (**D**) rats treated with thioacetamide + 250 mg/kg *A. muricata*, and (**E**) Rats treated with thioacetamide + 500 mg/kg *A. muricata*. (GA) Gross morphology, (H&E) Hematoxylin and Eosin stain 20x, (MT) Masson trichrome stain 20x.

### 3.3. Histopathological Examination of Liver Sections

The normal control group showed intact livers, and histological examination revealed ordinary cellular construction (Figure 2). However, the histology of the TAA control group disclosed severe liver injury as proved by the existence of inflammatory cell and mononuclear infiltrate, necrotic tissue, fibrosis and collagen production, the propagation of hepatic cells, bile ducts, and the vacuolation of cytoplasm. The regeneration of micro- and macro-nodules bound by fibrous connective tissues separated the liver into pseudo lobules (Figure 2).

*A. muricata* or silymarin-fed groups displayed significantly minor impairment in contrast to the widespread liver injuries initiated in the TAA control group. *A. muricata* or Silymarin prohibited inflammation, edema, mononuclear cell accumulation, necrotizing hepatocytes, and connective tissue fiber propagation which is produced by TAA.

Subsequently, liver architecture conserved their closely typical hepatic lobular construction. Outcomes verified the defensive roles of *A. muricata* or silymarin in contradiction to TAA-induced liver injury. A histopathology review of hepatic slices of rats nourished by *A. muricata* or silymarin exposed diminished grades of cirrhosis, nevertheless slighter necrotized regions, micronodules, and minor quantities of vacuoles in the cytoplasm and nucleic damage (Figure 2).

### 3.4. Immunohistochemistry Stain

The importance of *A. muricata* on cellular propagation and subsequent TAA-induced liver impairment was observed via the immunostaining of proliferating cell nuclear antigen (PCNA) in hepatic and spleen sections using an anti-PCNA antibody. Liver or spleen cells of the normal control group exhibited no PCNA discoloration, representing that there is no occurrence of cell renewal. On the contrary, hepatic or spleen cells in the hepatotoxic group had over-expressed PCNA and a raised mitotic index, signifying dissemination to the restoration of extensive injuries of liver and spleen structures made by TAA intoxication. *A. muricata* or silymarin groups had reduced hepatocytes and spleen cell renewal compared to the hepatotoxic control rats, for example, designated by a reduction in PCNA staining appearance and a substantial decrease in mitotic index, signifying the lower degree of necrotized hepatocytes and spleen cell propagation than of apoptosis (Figure 3, Table 3).

Thioacetamide (TAA)-induced liver injury and the importance of *A. muricata* were examined by the immunohistochemical staining of $\alpha$-smooth muscle actin ($\alpha$-SMA) and transforming growth factor-beta 1 (TGF-$\beta$1) expression in the liver parenchyma utilizing specific antibodies. The normal control group showed the down-regulation of $\alpha$-SMA staining, which indicates no occurrence of cell regeneration (Figure 4). On the contrary, the hepatotoxic control group had an outstanding $\alpha$-SMA appearance signifying up-regulation of these proteins with a higher level of hepatocyte fibrosis. The hepatotoxic control group elevated the mitotic figure index significantly, suggesting the proliferation of the regeneration of widespread hepatic damage induced by TAA. Rats fed 500 mg/kg *A. muricata* extract had reduced hepatic cell revitalization in comparison to the hepatotoxic control group, as indicated by $\alpha$-SMA and TGF-$\beta$1 appearance and a significant drop-off in the mitotic index. These results were comparatively similar to that of the silymarin-treated group. *A. muricata* resisted hepatocyte fibrosis by down-regulating $\alpha$-SMA expression. Rats fed with 250 mg /kg *A. muricata* extracts, however, exhibited mild to moderate expressions of $\alpha$-SMA and TGF-$\beta$1 within the hepatocytes with a significant decrease in the mitotic figure index, but not analogous to the silymarin-treated group. These results suggest that *A. muricata* had an estimable hepatoprotective effect by inhibiting the fibrosis of hepatocytes and ameliorating propagation.

**Liver**　　　　　　　　　**Spleen**

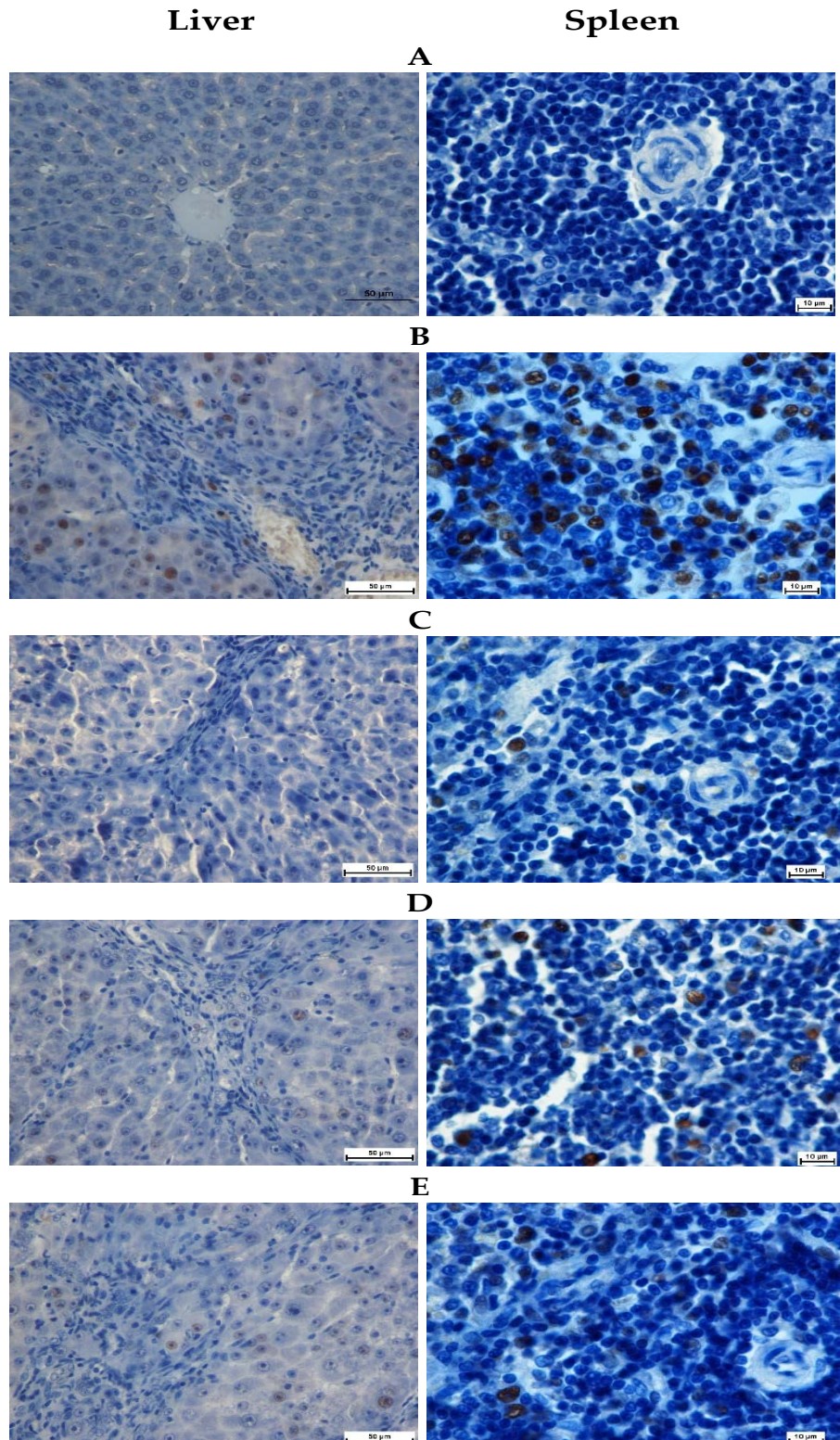

**Figure 3.** Immunohistochemistry stain (PCNA) of liver and spleen. Impacts of *A. muricata* extract on thioacetamide-induced liver and spleen injuries in experimental rats. (**A**) Normal control group, no PCNA staining (downregulation); (**B**) rats treated with thioacetamide (hepatotoxic group), PCNA expression hepatocyte nuclei (up-regulation); (**C**) rats treated with thioacetamide + silymarin, mild PCNA expression of hepatocytes nuclei (down-regulation); (**D**) rats treated with thioacetamide + 250 mg/kg *A. muricata*, mild to moderate expression PCNA-positive hepatocyte nuclei (down-regulation); and (**E**) rats treated with thioacetamide + 500 mg/kg *A. muricata* mild PCNA expression hepatocyte nuclei (down-regulation) (PCNA stain, magnification 40x).

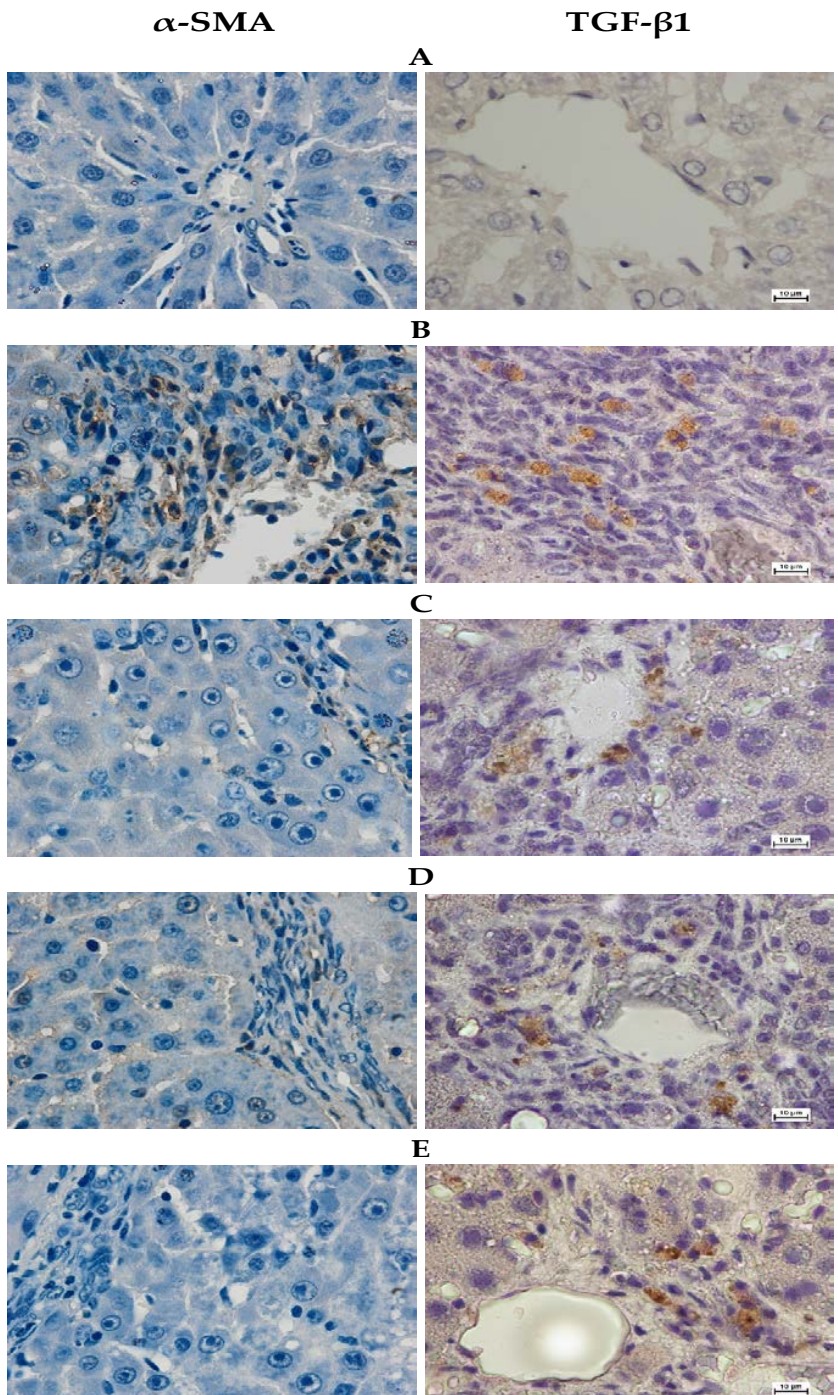

**Figure 4.** Immunohistochemistry: Effects of *A. muricata* extract on α-SMA staining in the liver. (**A**) Normal control group, no α-SMA expression (down-regulation); (**B**) rats treated with thioacetamide (hepatotoxic group), α-SMA expression of hepatocytes (up-regulation); (**C**) rats treated with thioacetamide + silymarin, mild α-SMA expression (down-regulation); (**D**) rats treated with thioacetamide + 250 mg/kg *A. muricata*, mild α-SMA expression (down-regulation), and (**E**) rats treated with thioacetamide + 500 mg/kg *A. muricata*, (down-regulation) (α-SMA stain, magnification 40x). TGF-β1 staining: (**A**) Normal control group, no TGF-β1 expression (down-regulation); (**B**) TAA-treated group, TGF-β1 expression of hepatocytes was seen (up-regulation); (**C**) TAA + Silymarin group, mild TGF-β1 expression (down-regulation), (**D**) TAA + 250 mg/kg *A. muricata*, mild TGF-β1 expression (down-regulation); (**E**) TAA + 500 mg/kg *A. muricata*, mild TGF-β1 expression (down-regulation) (TGF-β1 stain, magnification 40x).

### 3.5. Effect of A. muricata on Liver Antioxidant Enzyme and Oxidative Stress (Malondialdehyde (MDA)) Levels

Liver homogenates of the hepatotoxic group revealed pointedly condensed SOD and CAT activities in contrast to rats fed with *A. muricata*. *A. muricata* significantly restored the SOD and CAT activities by protecting the tissues from the hepatotoxic effects of TAA. However, MDA quantity upsurges in homogenized materials signify a subsidiary guide of lipid peroxidation. The MDA of liver tissue homogenate was significantly high in the TAA control group. However, the administration of this plant extract significantly decreased the level of MDA (Figure 5).

**Table 3.** Effect of *A. muricata* on the PCNA staining of liver and spleen sections and the mitotic index of the liver.

| Animal Groups | Liver PCNA Stain | Liver Mitotic Index | Spleen PCNA Stain |
|---|---|---|---|
| Normal Control | 0.00 | 0.00 | 0.00 |
| TAA Control (200 mg/kg) | 31 ± 4.77 [d] | 78.16 ± 3.71 [d] | 82.16 ± 4.70 [c] |
| Silymarin (50 mg/kg) + TAA | 1 ± 0.63 [a] | 20.16 ± 2.31 [a] | 2.18 ± 0.95 [a] |
| *A. muricata* (250 mg/kg) + TAA | 2 ± 0.89 [b] | 33.16 ± 2.63 [c] | 18.5 ± 1.87 [b] |
| *A. muricata* (500 mg/kg) + TAA | 3.16 ± 1.47 [c] | 26.16 ± 1.47 [b] | 2.16 ± 0.98 [a] |

Percentage of labeled hepatic cells per 1000 liver cells. Mean value ± SEM (*n* = 6). Values indicated by different superscripts within the same column are significantly different according to Tukey's honestly significant difference test at *p* < 0.05 significance level. There are no statistically significant changes among diverse groups.

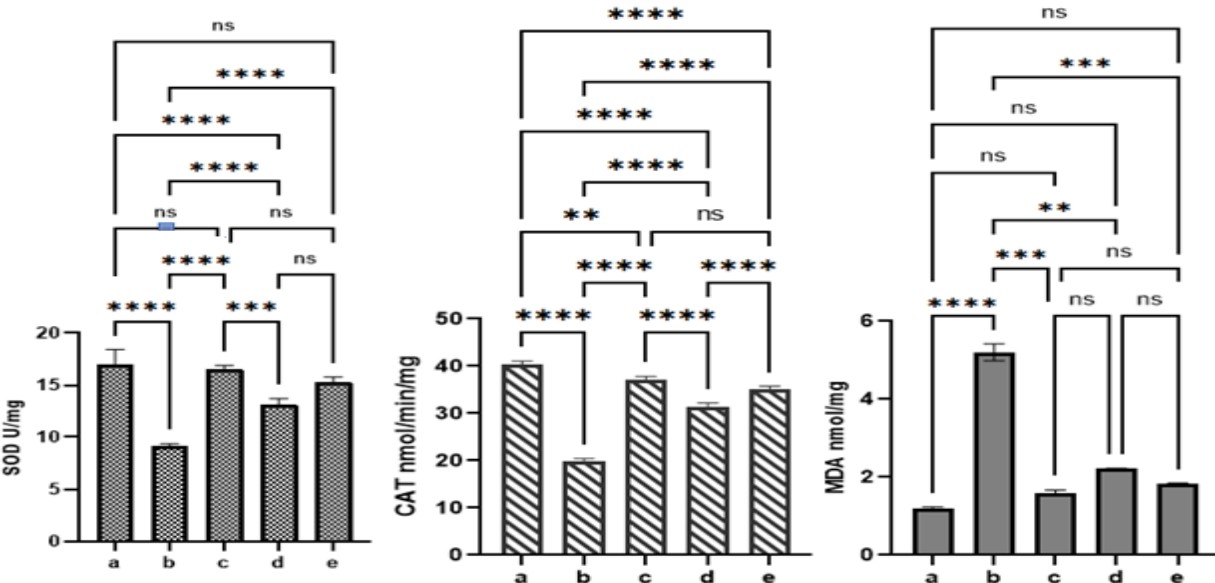

**Figure 5.** Effects of *A. muricata* extracts on antioxidant enzyme activities (SOD and CAT) and MDA levels in the liver. Data expressed as mean ± SEM. Means among groups (*n* = 6 rate/group) show the significant differences examined using one-way ANOVA and Tukey's post hoc multiple comparisons test. **** *p* < 0.0001, *** *p* < 0.001, ** *p* < 0.005 above columns specify significant differences from the normal control group and above lines for the differences between treated groups. ns *p* > 0.05 denotes non-significant differences. a. Normal Control, b. TAA Control (200 mg/kg), c. Silymarin (50 mg/kg) + TAA, d. *A. muricata* (250 mg/kg) + TAA, e. *A. muricata* (500 mg/kg) + TAA.

### 3.6. Effect of A. muricata on Biochemical Markers

The alkaline phosphatase (ALP), alanine aminotransferase (ALT), aspartate aminotransferase (AST), and bilirubin were significantly elevated in the TAA control group compared to the silymarin or *A. muricata* groups. However, albumin and total protein were significantly lesser in the TAA control group compared to rats who received silymarin or *A. muricata* (Table 4). The levels of liver function biomarkers were significantly decreased by *A. muricata* or silymarin feeding.

**Table 4.** Effects of *A. muricata* extracts on liver biochemical parameters in the serum of TAA-induced liver cirrhosis in rats.

| Animal Groups | ALP IU/L | ALT IU/L | AST IU/L | T. Bilirubin (μmol/L) | Protein g/L | Albumin g/L |
|---|---|---|---|---|---|---|
| Normal Control (10% Tween 20) | 72.33 ± 2.10 [a] | 39.2 ± 2.38 [a] | 62.15 ± 2.5 [a] | 12.05 ± 0.19 [a] | 69.54 ± 1.84 [a] | 24.27 ± 1.24 [a] |
| TAA control (200 mg/kg) | 251.2 ± 2.73 [d] | 174 ± 3.01 [d] | 254.0 ± 2.4 [d] | 5.04 ± 0.03 [d] | 42.90 ± 3.31 [c] | 14.81 ± 2.07 [e] |
| Silymarin (50 mg/kg) + TAA | 78.45 ± 1.0 [a] | 45.5 ± 1.21 [a] | 68.87 ± 1.6 [a] | 1.38 ± 0.07 [b] | 62.65 ± 1.62 [b] | 21.53 ± 2.04 [b] |
| *A. muricata* (250 mg/kg) + TAA | 96.75 ± 1.44 [c] | 83.42 ± 1.3 [c] | 87.66 ± 1.5 [c] | 2.32 ± 0.07 [c] | 58.89 ± 1.01 [b] | 16.56 ± 1.17 [d] |
| *A. muricata* (500 mg/kg) + TAA | 80.57 ± 2.13 [b] | 51.77 ± 2.5 [b] | 73.75 ± 2.0 [b] | 1.28 ± 0.7 [b] | 61.52 ± 1.9 [b] | 19.47 ± 0.99 [c] |

Mean value ± SEM (*n* = 6). Values indicated by different superscripts within the same column are significantly different according to Tukey's honestly significant difference test at a $p < 0.05$ significance level.

## 4. Discussion

In the present study, an oral acute toxicity trial of *A. muricata* on experimental rats discovered that there was no mortality during the test in animals fed with *A. muricata* extract relative to the normal control group, approving the harmless plant extract. Correspondingly, many studies by different researchers using diverse plant extracts have exhibited its harmlessness, and no sign of toxicity was noticed [6,44–48].

In the hepatotoxic group, there was a significant decrease in body weight accompanied by increased liver weight (hepatomegaly) compared to the normal control. Improved liver mass/body heaviness proportions in the hepatotoxic group could be attributed to the buildup of adipocytes and deterioration in the liver cells. Similarly, decreased body weight and increased liver weightiness in the hepatotoxic group were previously reported by different studies [1,4,37]. In contrast, *A. muricata*-fed rats reduced their liver weight to almost normal ranges compared to the hepatotoxic group. The decrease in the liver/body heaviness ratio perceived in *A. muricata* feeding may be owed to a decrease in hyperlipidemia [49] or might be due to reduced inflammation [50]. With the consistency of the results of our study, several researchers who used various plant extracts demonstrated a decrease in liver weight or body weight ratios compared to the hepatotoxic group [1,2,36,51].

Thioacetamide (TAA) injection produced liver cirrhosis in rats. However, rats fed with *A. muricata* could expressively hasten the retrieval of liver injury and significantly avoid the effects of TAA toxicity. The findings of this study are also in line with previous studies reported by several researchers using different medicinal plants in contradiction to TAA-induced liver impairment in rats [38,52–54]. Results of the current study showed condensed collagen production synthesis in *A. muricata* feeding, for example, Masson's trichrome stain. In agreement with the results of the present study, several studies which used different medicinal plant extracts showed a reduction in collagen fibers against TAA-induced liver cirrhosis [2,23,36,55].

The discovery of proliferating cell nuclear antigen (PCNA) using immune-histochemistry approaches the most collective means to understand the multiplying action of tissues. PCNA has lately been recognized using the polymerase S addition protein [56]. In this study, the normal liver and spleen control group or silymarin-fed group exhibited no significant PCNA discoloration, signifying the nonappearance of cell renewal. The over-regulation of PCNA appearance in hepatocytes or spleen cells was noticed in the hepatotoxic group, representing extensive propagation and probable efforts to restore tissue injury [37]. Alternatively, rats fed with silymarin or *A. muricata* showed significantly diminished cell proliferation quantities because of a decrease in PCNA staining.

In the TAA-induced hepatotoxic rats, TAA caused the production of reactive oxygen species (ROS) leading to the stimulation of hepatic satellite cells (HSC), a major factor for the ECM in chronic liver cirrhosis and the up-regulation of transforming growth factor-beta 1 (TGF-β1) and α-smooth muscle actin (α-SMA). The initiation of HSC comes with the cell propagation and upgrading of ECM production, and the occurrence of α-SMA in myofibroblasts. The results of the current study revealed that *A. muricata* supplementation caused the down-regulation of α-SMA compared to the hepatotoxic group, which showed a significant up-regulation of α-SMA. *A. muricata* significantly reduced the HSC activation

by decreasing the rate of ROS production. Similar results have been reported on the efficiency of medicinal plants in the down-regulation of α-SMA in TAA-induced liver cirrhosis [33,57,58].

In the liver tissue homogenate, superoxide dismutase (SOD) and catalase (CAT) enzymes were meaningfully condensed in the hepatotoxic group in contrast to the normal control group. In the hepatotoxic group, both enzymes were architecturally deteriorating due to free radical-resultant liver impairment [59]. In the meantime, *A. muricata* notably raised the concentration of serum CAT and SOD, defending the liver from the harmful effects of free radicals compared to the hepatotoxic group. Similar results have been reported previously by several researchers [7,18,36,53]. Lipid peroxidation is a commonly harmful process [60]. Malondialdehyde (MDA) levels elevated in tissue signify lipid peroxidation. A rise in MDA increases lipid peroxidation, causing injury to anti-oxidant protection apparatuses to avoid the development of extra free radicals [61]. The current investigation displayed that TAA produced a rise in MDA quantity, which was encouragingly reduced by *A. muricata* extract. Similar results have been earlier reported by previous studies [32,38]. Decreased hepatic SOD and CAT activities in the TAA control group may perhaps clarify increased MDA.

The hepatotoxic group was associated with a noticeable upsurge in blood markers alkaline phosphatase (ALP), alanine aminotransferase (ALT), aspartate aminotransferase (AST), and bilirubin levels. The elevation in serum liver biomarkers reflects hepatocellular injury. These values were significantly reduced, to approximately close to normal values in *A. muricata*-fed groups. Comparable explanations of serum liver indicator enhancement by using different medicinal plants were formerly informed by previous studies [32,53]. The hepatoprotective action may be because of its effect on cell leakage and damage to the hepatic cell sheath. TAA is stated to hinder RNA's drive from the nucleus to the cytoplasm, initiating casing injury, which results in an increase in serum liver indicators [38]

In our research, the total protein and total albumin quantities in serum were distinctly reduced in the hepatotoxic group. However, silymarin or *A. muricata* feeding groups bring these proteins back to a nearly normal level. Similarly, various researchers showed that rats who were treated with silymarin or several plant extracts showed normal levels of albumin and protein [27,31,37].

## 5. Conclusions

Based on the results of the present study, *A. muricata* revealed an expressively hepatoprotective effect in the inhibition of TAA-induced hepatic damage in rats, as shown by macroscopic appearance, histology, immunohistochemistry, and biochemical liver parameters. *A. muricata* dramatically increased the serum activity of CAT and SOD and significantly attenuated hepatic MDA. The extracts of *A. muricata* effectively prevent TAA-induced liver cirrhosis by the marked downregulation of hepatic PCNA, TGF-β1, and α-SMA expression. The defensive result of *A. muricata* against TAA-induced hepatotoxicity could be due to its capacity to avoid hepatocyte propagation, decrease oxidative stress and lipid peroxidation, and its antioxidant and free radical scavenger properties.

**Author Contributions:** Conceptualization, M.A.A., S.H.S. and I.A.A.I.; data curation, M.A.A.; formal analysis, A.A.J.; investigation, S.H.S. and M.A.A.; methodology, S.H.S. and M.H.A.-M.; software, S.H.S. and A.R.A.; supervision, M.A.A.; visualization, M.H.A.-M. and S.H.S.; writing—original draft, M.A.A. and S.H.S.; writing—review and editing, M.A.A., S.H.S., A.A.J., I.A.A.I. and M.H.A.-M. All authors have read and agreed to the published version of the manuscript.

**Funding:** This research was funded by Deanship of Scientific Research at Umm Al-Qura University/Saudi Arabia, grant number 22UQU4331277DSR02.

**Institutional Review Board Statement:** Not applicable.

**Informed Consent Statement:** Not applicable.

**Data Availability Statement:** Data supporting the current study are available on request.

**Acknowledgments:** The authors would like to thank the Deanship of Scientific Research at Umm Al-Qura University for supporting this work by Grant Code: (22UQU4331277DSR02).

**Conflicts of Interest:** The authors declare no conflict of interest.

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
