# Peer review of "Histopathological Evaluation of Annona muricata in TAA-Induced Liver Injury in Rats"

_processes, doi:10.3390/pr10081613_

Round 1

Reviewer 1 Report

Dear authors,

The manuscript entitled “Histopathological evaluation of Annona muricata in TAA-induced liver injury in rats” is very interesting research, the authors evaluate the toxicity of the extract and the hepatoprotective activity in a TAA model in rats.

I have some observations in the manuscript.

-          In the Abstract, in paragraph 24, the author mention “Sprague drawly”, this need to be corrected in all the manuscript.

-          In the introduction there are need to be more explained the incidence of liver fibrosis, and if there are a pharmacological treatment.

-          Why A. muricata was selected to be evaluated?

-          The applicative uses in liver cirrhosis models of annonaceae family must be explained.

-          Is there another Annona species that are used to treat liver cirrhosis?

-          There are studies that demonstrate which metabolite is the responsible of the activity on liver cirrhosis model?

I think that these points can be used to explore more on this section.

-          In the paragraph 53 to 66 until reference [6, 18, 26-29] should be in the discussion section.

-          In the paragraph 71, I think that the sentence “This investigation” can be more useful than “This education”.

-          Despite the abbreviations are mentioned in the abstract, I suggest that all the abbreviations need to be explained in the main text for the first time, such as TAA, MDA, etc.

-          In paragraph 86 the author mentions a daily treatment for two months and in paragraph 95 mention that the treatment is going to be administered during eight weeks, I know that is the same time of treatment. But in this case, I suggest being homogeneous in the redaction.

-          The animal section in material and methods must be explained before acute toxicity study, also, the mean of SD abbreviation is not mentioned in the text.

-          The acute toxicity procedure is explained in any guideline or something similar? Like the OECD tests guidelines for chemicals, or the author only based on articles to carry out these assays. Why are the doses of 2000 and 5000 mg/kg used? are ethically correct the use of that doses? This must be supported by a guideline.

-          The tables must be checked, “Animals’ groups” is not correct. Also, A. muricate is not correct. If the dose is expressed in grams/kilogram it must be written as “g/kg”, not “gm/kg”. The same with the other units, in some tables are written as mmol/L and in other as mmol/l, all the units must be checked.

-          The figure 1 must be corrected, there are not a figure D, E and F.

-          “A. muricate” must be corrected in all the text.

-          In table 2 grams must be expressed as (g).

Additionally, an extensive english edition is required 

Author Response

The manuscript was checked and corrected. 

Reviewer 2 Report

Overall I think thiw could be an interesting article. However it is hampered by the extremely low quality of written english. I liked your pictures and tables, but before your article can even be eligible for a proper review written english quality must be greatly improved

Author Response

The Manuscript linguistically corrected

Round 2

Reviewer 1 Report

Se han hecho todas las sugerencias.

Sugiero aceptar el manuscrito en su forma actual.

Author Response

Are all the cited references relevant to the research?

Yes, all the cited references are relevant to the research

Is the research design appropriate

Yes, the research design is appropriate as shown by the sequence of headings in materials and methods

Are the results clearly presented?

Yes, the results are clearly presented as shown by obvious Tables and figures

Reviewer 2 Report

Overall much improved manuscript.

1. English still need improvement in some parts (t.ex. line 68 "abundant liver disorders" should be replaced with "a variety of liver disorders", sentence from lines 71-74 don't make any sense, in line 79 "training" should be replaced with "study" etc)

2. The meaning of each abbreviation should be mentioned at the first time they are mentioned in the main text and only then

3.  Lipid peroxidation in line 44 should be replaced with metabolic-associated fatty liver disease, since you are talking about diseases and not pathogenetic mechanisms leading to liver fibrosis

4. Likewise, food poisoning should be ommited since it does not lead to chronic liver fibrosis

5. In line 53 references should be given with A. muricata serving as a medicine for various liver diseases

6. In line 137 per is lacking the os (it should be per os)

7. The sentence starting in line 204 is not a sentence since it is lacking a verb; I think it should be part of the previous sentence

8. In line 208, I don't really understand what "diverse collections" means

9. In line 271 "expressively" doesn't make any sense

10. It would be interesting to see if there were any differences between silymarin and A. muricata in your rats

Author Response

  1. English still need improvement in some parts (t.ex. line 68 "abundant liver disorders" should be replaced with "a variety of liver disorders", sentence from lines 71-74 don't make any sense, in line 79 "training" should be replaced with "study" etc)

-Checked and corrected

  1. The meaning of each abbreviation should be mentioned at the first time they are mentioned in the main text and only then

-Corrected

  1. Lipid peroxidation in line 44 should be replaced with metabolic-associated fatty liver disease, since you are talking about diseases and not pathogenetic mechanisms leading to liver fibrosis

-Corrected

  1. Likewise, food poisoning should be omitted since it does not lead to chronic liver fibrosis

-Omitted

  1. In line 53 references should be given with A. muricata serving as a medicine for various liver diseases

All parts of A. muricata, counting the bark, fruit seeds, leaves and root are utilized in usual medications in the tropics including liver complications.

  1. In line 137 per is lacking the os (it should be per os)

-Corrected

  1. The sentence starting in line 204 is not a sentence since it is lacking a verb; I think it should be part of the previous sentence.

-Replaced verb

  1. In line 208, I don't really understand what "diverse collections" means

-A diverse collection means various groups

  1. In line 271 "expressively" doesn't make any sense

- Omitted

  1. It would be interesting to see if there were any differences between silymarin and A. muricata in your rats

Rat’s body weight and Liver index were significantly higher in silymarin group compared to A. muricata, whereas no significant differences in liver weight between silymarin group and A. muricata groups

There were no significant differences in the gross appearance of the liver between the silymarin group and A. muricata groups

There were no significant differences in histopathological examination of liver sections between the silymarin group and 500mg/kg A. muricata group

There were significant differences in immunohistochemistry stain (PCNA) examination of liver and spleen sections between the silymarin group and A. muricata groups. Silymarin significantly showed down-regulation of PCNA, α-SMA and TGF-β1 staining compared to A. muricata groups.